# 5 Years of GOSAT-2 Retrievals with RemoTeC: XCO<sub>2</sub> and XCH<sub>4</sub> Data Products with Quality Filtering by Machine Learning

Andrew Gerald Barr<sup>1</sup>, Jochen Landgraf<sup>1</sup>, Mari Martinez-Velarte<sup>1</sup>, Mihalis Vrekoussis<sup>2,3</sup>, Ralf Sussmann<sup>4</sup>, Isamu Morino<sup>5</sup>, Kimberly Strong<sup>6</sup>, Minqiang Zhou<sup>7</sup>, Voltaire A. Velazco<sup>8</sup>, Hirofumi Ohyama<sup>9</sup>, Thorsten Warneke<sup>3</sup>, Frank Hase<sup>10</sup>, and Tobias Borsdorff<sup>1</sup>

**Correspondence:** Andrew Gerald Barr (a.g.barr@sron.nl)

**Abstract.** Accurately measuring greenhouse gas concentrations to identify regional sources and sinks is essential for effectively monitoring and mitigating their impact on the Earth's changing climate. In this article we present the scientific data products of  $XCO_2$  and  $XCH_4$ , retrieved with RemoTeC, from the Greenhouse Gases Observing Satellite-2 (GOSAT-2), which span a time range of five years. GOSAT-2 has the capability to measure total columns of  $CO_2$  and  $CH_4$  to the necessary requirements set by the Global Climate Observing System (GCOS), who define said requirements as accuracy < 10 ppb and < 0.5 ppm for  $XCH_4$  and  $XCO_2$  respectively, and stability of < 3 ppb yr<sup>-1</sup> and < 0.5 ppm yr<sup>-1</sup> for  $XCH_4$  and  $XCO_2$  respectively.

Central to the quality of the XCO<sub>2</sub> and XCH<sub>4</sub> datasets is the post-retrieval quality flagging step. Previous versions of RemoTeC products have relied on threshold filtering, flagging data using boundary conditions from a list of retrieval parameters. We present a novel quality filtering approach utilising a machine learning technique known as Random Forest Classifier (RFC) models. This method is developed under the European Space Agency's (ESA) Climate Change Initiative+ (CCI+) program and applied to data from GOSAT-2. Data from the Total Carbon Column Observing Network (TCCON) are employed to train the RFC models, where retrievals are categorized as good or bad quality based on the bias between GOSAT-2 and TCCON measurements. TCCON is a global network of Fourier transform spectrometers that measure telluric absorption spectra at infrared wavelengths. It serves as the scientific community's standard for validating satellite-derived XCO<sub>2</sub> and XCH<sub>4</sub> data. Our results demonstrate that the machine learning-based quality filtering achieves a significant improvement, with data yield increasing by up to 85% and RMSE improving by up to 30%, compared to traditional threshold-based filtering. Furthermore, inter-comparison with the TROPOspheric Monitoring Instrument (TROPOMI) indicates that the quality filtering RFC models generalise well to the full dataset, as the expected behaviour is reproduced on a global scale.

<sup>&</sup>lt;sup>1</sup>Earth science group, SRON Netherlands Institute for Space Research, Niels Bohrweg 4, 2333 CA Leiden, The Netherlands

<sup>&</sup>lt;sup>2</sup>Climate and Atmosphere Research Center (CARE-C), The Cyprus Institute, Nicosia, Cyprus

<sup>&</sup>lt;sup>3</sup>Institute of Environmental Physics, University of Bremen, Bremen, Germany

<sup>&</sup>lt;sup>4</sup>Karlsruhe Institute of Technology (KIT), IMK-IFU, Garmisch-Partenkirchen, Germany

<sup>&</sup>lt;sup>5</sup>Satellite Remote Sensing Section and Satellite Observation Center, Earth system Division, National Institute for Environmental Studies (NIES), Onogawa 16-2, Tsukuba, Ibaraki 305-8506, Japan

<sup>&</sup>lt;sup>6</sup>Department of Physics, University of Toronto MP710A, 60 St. George Street, Toronto, ON, M5S 1A7, Canada

<sup>&</sup>lt;sup>7</sup>Institute of Atmospheric Physics, Chinese Academy of Sciences, Beijing 100029, China

<sup>&</sup>lt;sup>8</sup>Deutscher Wetterdienst (DWD), Meteorological Observatory Hohenpeissenberg, 82383 Hohenpeissenberg, Germany

<sup>&</sup>lt;sup>9</sup>Earth System Division, National Institute for Environmental Studies, Tsukuba, Japan

<sup>&</sup>lt;sup>10</sup>Institute of Meteorology and Climate Research (IMK-ASF), Karlsruhe Institute of Technology (KIT), Karlsruhe, Germany

Low systematic biases are essential for extracting meaningful fluxes from satellite data products. Through TCCON validation we find that all data products are within the breakthrough bias requirements set, with RMSE for XCH<sub>4</sub> <15 ppb and XCO<sub>2</sub> <2 ppm. We derive station-to-station biases of 4.2 ppb and 0.5 ppm for XCH<sub>4</sub> and XCO<sub>2</sub> respectively, and linear drift of 0.6 ppb  $yr^{-1}$  and 0.2 ppm  $yr^{-1}$  for XCH<sub>4</sub> and XCO<sub>2</sub> respectively.

For XCH<sub>4</sub>, GOSAT-2 and TROPOMI are highly correlated with standard deviations less than 18 ppb and globally averaged biases close to 0 ppb. The inter-satellite bias between GOSAT and GOSAT-2 is significant, with an average global bias of -15 ppb. This is comparable to that seen between GOSAT and TROPOMI, consistent with our findings that GOSAT-2 and TROPOMI are in close agreement.

### 1 Introduction

Anthropogenic emissions of greenhouse gases (GHGs) such as carbon dioxide (CO<sub>2</sub>) and methane (CH<sub>4</sub>) over the last century have led to the rapid rise of concentrations of GHGs in the atmosphere (Figure 1.4, IPCC AR6 2021, Tans and Keeling (2020), Cross-Chapter Box 5.2 IPCC AR6 2021). The effect of such changes in atmospheric composition has a clear correlation with the change of climate variables - such as global sea surface temperature anomaly or sea level - with CO<sub>2</sub> increase over preindustrial levels directly proportional to global mean surface temperature anomaly, relative to 1850-1900 (Figure 1.6, IPCC AR6 2021). Indeed, the emergence of trend in climate variables above the natural year-to-year variability has been firmly established (Banks and Wood, 2002; Giorgi and Bi, 2009; Lyu et al., 2014; Hawkins and Sutton, 2012; IPCC AR5, 2014; Tebaldi and Friedlingstein, 2013), on a global scale as well as regional ones (Mahlstein et al., 2011; Hawkins et al., 2020; Rohde and Hausfather, 2020). The ramifications of a warming climate are serious with significant negative implications affecting the entire globe.

Satellite retrievals of concentrations of CO<sub>2</sub> and CH<sub>4</sub>, or rather column-averaged dry air mole fractions, denoted XCH<sub>4</sub> and XCO<sub>2</sub>, play an essential role in monitoring the changing climate, as these variables can be used alongside inverse modelling of surface fluxes to estimate uptake and emission of GHG surface fluxes (Bergamaschi et al., 2009; Chevallier et al., 2007, 2005; Meirink et al., 2006; Metz et al., 2023). In particular satellite measurements that are sensitive to near-surface variations in GHG concentrations are essential, and tight requirements are necessary to accurately calculate fluxes and so quantify emissions. The Global Carbon Observing System (GCOS) has classified measurements of CO<sub>2</sub> and CH<sub>4</sub> columns as Essential Climate Variables (ECVs), and defines requirements as being accurate enough to be able to determine sources and sinks on regional scales (GCOS, 2016). To this end, ESA's Climate Change Initiative (CCI) seeks to achieve this with the GHG-CCI+ project in which ECVs of CO<sub>2</sub> and CH<sub>4</sub> columns are delivered globally.

Particular emphasis is placed on systematic biases in satellite data, such as the change in bias over time, of which the requirements on XCO<sub>2</sub> and XCH<sub>4</sub> are less than 0.5 ppm yr<sup>-1</sup> and 3 ppb yr<sup>-1</sup> respectively (GCOS, 2016). Furthermore, the station-to-station bias of sites, or accuracy, from the The Total Carbon Column Observing Network (TCCON), defined as the standard deviation of all station biases, should be less than 0.5 ppm for XCO<sub>2</sub> and less than 10 ppb for XCH<sub>4</sub> (GCOS, 2016).

The Japan Aerospace Exploration Agency (JAXA) operated satellite GOSAT-2 (Greenhouse Gases Observing Satellite-2) has onboard the TANSO-FTS-2 instrument (Thermal And Near infrared Sensor for carbon Observation-Fourier Transform Spectrometer-2), which operates in the near-infrared (NIR), short-wave infrared (SWIR) bands, as well as the thermal infrared. TANSO-FTS-2 has sufficient sensitivity to measure regional sources and sinks of GHGs, and provides calibrated and geolocated Earthshine radiance spectra (level-1B data) in the aforementioned wavelength regimes, with 10 km circular ground pixels covering the globe every 6 days in sun-synchronous orbit (Suto et al., 2021; Imasu et al., 2023). TANSO-FTS-2 has an intelligent pointing system, allowing better coverage than its predecessor GOSAT. Also onboard is the dedicated cloud imager TANSO-CAI-2 (Thermal And Near infrared Sensor for carbon Observation-Cloud and Aerosol Imager-2) (Kuze et al., 2009, 2016; Yoshida et al., 2012).

GOSAT was the first dedicated GHG observing satellite, and has been used in a wide variety of scientific studies relevant to CO<sub>2</sub> and CH<sub>4</sub> since 2009 (Butz et al., 2011; Schepers et al., 2012; Parker et al., 2020; Taylor et al., 2022). Trace gas column-averaged dry air mole fractions, also referred to as the level 2 product, can be extracted from the level 1B data through a retrieval (see section 3). Crucial for the carbon cycle, fluxes of CO<sub>2</sub> have been inferred from level 2 data on regional scale (Chevallier et al., 2009; Basu et al., 2013; Detmers et al., 2015) as well as global scales (Turner et al., 2015; Jiang et al., 2021; Kou et al., 2023). Also for CH<sub>4</sub>, global flux estimates and emissions (Maasakkers et al., 2019; Zhang et al., 2021) have been derived from GOSAT measurements, and also compared to the TROPOspheric Monitoring Instrument (TROPOMI) (Liang et al., 2023) and airborne in-situ measurements (Tadić et al., 2012).

In addition to the data we present in this article, two other XCO<sub>2</sub> and XCH<sub>4</sub> data products are available from GOSAT-2 (Noël et al., 2021, 2022; Yoshida et al., 2023). Noël et al. (2022) also present results for XH<sub>2</sub>O, as well as XCO and XN<sub>2</sub>O from GOSAT-2. Zadvornykh et al. (2023) investigated the retrieval of HDO/H<sub>2</sub>O ratio combining the NIR and thermal infrared bands, and Malina et al. (2018) presented a proof of concept study on retrieving <sup>13</sup>CH<sub>4</sub> from GOSAT-2. Ohyama et al. (2024) calculated emissions estimates from enhancement ratios of CO<sub>2</sub>, CH<sub>4</sub> and CO using inverse modelling and compared to emission inventories. Janardanan et al. (2025) compared flux inversions of CH<sub>4</sub> from GOSAT and GOSAT-2 across 2019-2022, finding regional differences in the emission estimates related to differences in the level 2 products, however assimilated GOSAT-2 XCH<sub>4</sub> data were not bias-corrected.

TCCON provides the most robust measure of the accuracy of total columns of GHGs measured by satellites (Wunch et al., 2010, 2011, 2015), and is widely used as the conventional validation for XCH<sub>4</sub> and XCO<sub>2</sub> retrievals (e.g. Dils et al. (2014); Malina et al. (2022)). It is a global network of Fourier transform spectrometers that observe, among others, XCO<sub>2</sub> and XCH<sub>4</sub> with a root mean square error (RMSE) on mole fractions of 0.15 % and 0.2 % respectively (Toon et al., 2009), for the GGG2014 release. Depending on the site, these measurements are scaled to aircraft or balloon-borne measurements for calibration (Washenfelder et al., 2006; Deutscher et al., 2010; Karion et al., 2010; Messerschmidt et al., 2011; Geibel et al., 2012) and measurements of vertical profiles can vary per site. TCCON measures direct sunlight and can therefore only be performed under clear-sky conditions hampering coverage of the time-series.

## 2 Data products and Input Data

In this article we present the novel level 2 GOSAT-2 scientific data products developed by SRON the Netherlands Institute for Space Research. XCO<sub>2</sub> and XCH<sub>4</sub> are retrieved deploying the RemoTeC retrieval algorithm, and are processed within the GHG-CCI+ project (Dils et al., 2014; Buchwitz et al., 2015). RemoTeC uses two different retrieval approaches which we discuss further in section 3. From the two configurations available to RemoTeC, three column-averaged dry air mole fractions products are produced. GOSAT-2 has been operational since February 2019. The data products discussed here cover the time range of the first observations until the end of 2023. Data products are available from the ESA Climate Office, under version v2.0.3 in Climate Data Research Package 9 (CDRP9) <sup>1</sup>. More information about the three SRON GOSAT-2 data products can be found in the Algorithm Theoretical Basis Documents (ATBDs) (Barr et al., 2024a, b) and Product User Guides (PUGs) (Barr et al., 2024c, d).

The SRON GOSAT-2 data products are generated from calibrated TANSO-FTS-2 L1B data from v210.210 for 2019 until June 2023, made available by the National Institute for Environmental Studies (NIES). For the second half of 2023 we used L1B from v220.220. Instrument line shape (ILS) information is taken from Suto et al. (2021).

A pre-processing step brings meteorological data, surface data and satellite data together before the retrieval is run. Meteorological input data are taken from the ECMWF ERA5 reanalysis product on 137 altitude layers and a  $0.75^{\circ} \times 0.75^{\circ}$  latitude/longitude grid (Hersbach et al., 2020). Surface information was taken from the extended Shuttle Radar Telemetry Mission (SRTM) digital elevation map. The model used, DEM3, has global coverage at 90 meter spatial resolution<sup>2</sup>, extending the original SRTM which is limited to latitudes of 56°S to 60°N. The solar reference spectra used for the retrieval is compiled from the full resolution spectrum of Kurucz (1994).

Absorption cross sections come from the HITRAN 2008 database for spectroscopic parameters (Rothman et al., 2009). Apriori column density profiles for  $CO_2$  and  $CH_4$  we take from TM5 (Huijnen et al., 2010) and TM4 (Meirink et al., 2006) model simulations respectively. For the  $XCH_4$  Proxy product,  $XCO_2$  data is used from the CAMS global inversion-optimised greenhouse gas concentrations of Chevallier (2010). These are surface air-sample instantaneous 3 hourly mean columns on  $1.9^{\circ} \times 3.75^{\circ}$  grids.

## 3 Retrieval

RemoTeC is a retrieval algorithm developed for the retrieval of trace gas column-averaged dry air mole fractions from measured level 1B radiance spectra in the NIR and SWIR bands. It has been used extensively for the retrieval of trace gases from GOSAT observations to produce the SRON XCH<sub>4</sub> and XCO<sub>2</sub> data products (Butz et al., 2009, 2010), as well as the operational products of TROPOMI (Hu et al., 2016, 2018; Lorente et al., 2021) and SCHIAMACHY (Frankenberg et al., 2005, 2011; Dils et al., 2006, 2014). Below we outline the retrieval approach. The same approach is also used to generate the GOSAT-2 data products.

<sup>1</sup>https://catalogue.ceda.ac.uk/

<sup>&</sup>lt;sup>2</sup>http://www.viewfinderpanoramas.org/dem3.html

An XCH<sub>4</sub> Full Physics product is obtained using the scattering forward model, and an XCO<sub>2</sub> Full Physics product is extracted from the same retrieval. Furthermore another XCH<sub>4</sub> product (the Proxy product) is obtained with the non-scattering forward model. For the Full Physics approach, light scattering by cirrus and aerosol particles is accounted for in the forward model. For the Proxy retrieval, scattering is neglected and hence atmospheric scattering properties do not need to be calculated (Butz et al., 2009).

An example of a single, typical GOSAT-2 measurement is shown in Figure 1 in which the spectral fits per band are presented for a high quality, cloud free scene in the Full Physics retrieval setup. The SWIR-1 window is split into two to retrieve total columns of CO<sub>2</sub> and CH<sub>4</sub> separately from band 2a and band 2b, respectively.

## 3.1 Forward Model

Both the scattering and non-scattering forward models have the same general concept in common which we outline here. The atmospheric state vector, x, is related to the measurement vector, y, through a forward model, F, which in the following equation:

$$y = F(x, b) + \epsilon_y + \epsilon_F \tag{1}$$

where  $\epsilon_y$  and  $\epsilon_F$  are the error contributions from the measurement noise and forward model respectively, and b is the ancillary vector containing parameters that are not retrieved. In order that the retrieval can be solved iteratively, the forward model must be linearised. For iteration step n the linearised forward model is approximated by:

$$130 \quad F(\boldsymbol{x}, \boldsymbol{b}) = F(\boldsymbol{x}_{\boldsymbol{n}}, \boldsymbol{b}) + K(\boldsymbol{x} - \boldsymbol{x}_{\boldsymbol{n}})$$

where  $x_n$  is the state vector for the n-th iteration step and K is the Jacobian matrix at position  $x_n$  defined by:

$$K = \frac{\partial F}{\partial x} \tag{3}$$

The inversion method optimises the state vector x with respect to the measurement y after applying the forward model F to x. The inversion method is based on the Tikhonov regularization scheme (Philips, 1962; Tikhonov, 1963; Hasekamp and Landgraf, 2005). Regularisation is required because the inverse problem is ill-posed (the measurements y typically contains insufficient information to retrieve all state vector elements independently). The inverse algorithm finds x by minimising the cost function that is the sum of the least-squares cost function and a side constraint weighted by the regularisation parameter y according to

$$\hat{x} = \underset{min}{x} (||S_y^{1/2}(F(x) - y)||^2 + \gamma ||W(x - x_a)||^2)$$
(4)

**Figure 1.** *left panels:* A single GOSAT-2 measurement for the near-infrared (band 1), SWIR-1 (band 2) and SWIR-3 (band 3) spectral windows (blue), along with the converged model (orange). *right panels:* The difference between the measurement and model shown on the left panels. The noise level is indicated by the black dashed lines.

where  $S_y$  is the diagonal measurement error covariance matrix, which contains the noise estimate,  $x_a$  is an a priori state vector, and W is a diagonal weighting matrix.

## 3.2 Proxy Approach

The Proxy approach is based on a non-scattering retrieval, thus the runtime of processing is around a factor of 4 faster than the Full Physics retrieval. Furthermore, many of the errors in the retrieval, including those due to aerosol, cancel out (Butz et al., 2009; Schepers et al., 2012) following the equation:

$$XCH_4 = \frac{[CH_4]}{[CO_2]}XCO_{2,model}$$
(5)

which determines XCH<sub>4</sub> from the retrieved total columns [CH<sub>4</sub>] and [CO<sub>2</sub>]. Here, the assumption is that the light path modification by scattering particles such as aerosols is the same for CH<sub>4</sub> and CO<sub>2</sub> (Schepers et al., 2012). [CH<sub>4</sub>] and [CO<sub>2</sub>] are total columns retrieved from SWIR-1 at 1.6  $\mu$ m, and XCO<sub>2,model</sub> is the total column dry air mixing ratio of CO<sub>2</sub> from an atmospheric model, on the same grid as GOSAT-2 observations. The main source of uncertainty in this approach is therefore XCO<sub>2,model</sub>, thus the accuracy of the XCH<sub>4</sub> Proxy product is limited by the accuracy of the XCO<sub>2</sub> model.

The state vector of the Proxy retrieval contains CO<sub>2</sub> and CH<sub>4</sub> sub-columns in 12 vertical layers, H<sub>2</sub>O total column, Lambertian surface albedo, first order spectral dependence of surface albedo, an intensity offset and first order spectral shifts of Earth and Sun radiancies (Barr et al., 2024b). We do not retrieve any information about the ILS such as shift or stretch parameters.

### 155 3.3 Full Physics Approach

160

The Full Physics retrieval uses a three-window approach retrieving information from the NIR, SWIR-1 and SWIR-3 bands. The treatment of aerosol in the Full Physics approach leads to more accurate retrieved total columns of trace gases, however the radiative transfer calculations are computationally expensive. The state vector of the Full Physics retrieval is the same as for the Proxy retrieval with additional parameters related to aerosol properties. For a full description of the state vector and the priors see Section 3.3 of Barr et al. (2024a).

Aerosols are characterised by three parameters which relate to the aerosol column and size distribution of particles. The height distribution is approximated as a Gaussian function of centre height,  $z_{aer}$  and width  $\omega_0$ :

$$h(z_k) = N \cdot \exp\left[-\frac{4ln(z_k - z_{aer})^2}{(2\omega_0)^2}\right] \tag{6}$$

Here, N is the total amount of particles and  $z_k$  is the layer height. The size distribution is parameterised by a power law function following:

$$n(r) = \begin{cases} A & \text{for } r \le r_1 \\ A(r/r_1)^{-\alpha} & \text{for } r_1 < r \le r_2 \\ 0 & \text{for } r > r_2 \end{cases}$$

$$(7)$$

where  $r_1$  = 0.1  $\mu$ m,  $r_2$  = 10  $\mu$ m and the constant A is determined from normalisation of the size distribution. N,  $\alpha$  and  $z_{aer}$  are included in the Full Physics state vector.

#### 3.4 Bias Correction

A bias correction is applied post-retrieval to XCO<sub>2</sub> and XCH<sub>4</sub> using TCCON as a truth, for which we use the GGG2020 release (Laughner et al., 2023). For land retrievals, the bias correction of RemoTeC is a simple empirical relation between XCH<sub>4</sub> and the retrieved albedo at 1600 nm, defined by:

$$X_{\text{corr}} = X_{\text{ret}}(a + b\alpha)$$
 (8)

where  $X_{\text{ret}}$  and  $X_{\text{corr}}$  are the bias corrected and retrieved concentrations respectively,  $\alpha$  is the retrieved albedo at 1600 nm and a and b are determined such that the difference with TCCON is minimised.

For the retrieval with the Proxy approach, the bias correction is all contained in the a variable of the fit (equation 8), therefore it is purely a constant bias correction, whereas the Full Physics approach has more contribution from the linear part of the fit, captured by the b parameter. This can be understood as confusion between albedo and aerosol effects in the retrieval, both of which lead to large scale wavelength features in the spectrum.

### 180 4 Quality Filtering

A key step in the retrieval process is the post-processing quality flagging. Data from GOSAT are flagged using a selection of retrieval parameters, such as signal-to-noise ratio or chi-squared, and any data that do not lie within a specified range of these parameters are flagged as bad quality. This method offers a binary quality flag and is described further in section 4.1.

Given the rapidly growing capabilities of machine learning techniques, algorithms such as random forest classifiers (RFCs) provide a much more promising way of filtering satellite global data products. We have applied such a flagging technique to the GOSAT-2 data (see section 4.2). The quality flag of GOSAT-2 takes the form of a quality assurance (QA) value that ranges between 0 and 1, with 0 corresponding to the best quality data. Therefore, users should quality filter their data by taking QA values less than, or equal to, the desired value.

#### 4.1 Threshold Criteria Approach

Extensive investigations have been conducted to identify effective retrieval parameters, or combinations of parameters, that are correlated with the quality of XCH<sub>4</sub> or XCO<sub>2</sub> and that can be used to flag bad data, while at the same time maximising the

Table 1. List of threshold conditions to quality filter GOSAT-2 data. The filters marked with an asterisk do not apply to the Proxy product.

| Filter | Description                                                | criteria                                                                      |
|--------|------------------------------------------------------------|-------------------------------------------------------------------------------|
| 1      | $\chi^2$ of spectral fit                                   | $\chi^2 < 12.0$                                                               |
| 2      | number of iterations $N_{ m iter}$                         | $N_{ m iter} < 31$                                                            |
| 3      | signal-to-noise ratio (SNR) at band continuum              | SNR > 50                                                                      |
| 4      | variance $\sigma_{ m surf}$ of surface elevation           | $\sigma_{ m surf}$ < 100 m                                                    |
| 5      | solar zenith angle (SZA)                                   | $SZA < 75^{\circ}$                                                            |
| 6*     | aerosol optical thickness ( $	au_{ m aer}$ ) in NIR window | $	au_{ m aer} < 1.0$                                                          |
| 7*     | aerosol size parameter $r_{\rm eff}$                       | $3 < r_{\rm eff} < 6$                                                         |
| 8*     | aerosol layer height $z_{ m aer}$                          | $0 < z_{\rm aer} < 10,\!000 \; {\rm m}$                                       |
| 9*     | aerosol parameter $\omega$                                 | $0 < \omega < 3e4$                                                            |
| 10     | blended albedo $A_{\rm bld}$                               | $0 < A_{\rm bld} < 1.4$                                                       |
| 11     | cirrus radiance signal $I_{ m cir}$                        | $0 < I_{\rm cir} < 2.0 \cdot 10^{-9}  [{\rm W  cm}/({\rm m}^2  {\rm s  str}]$ |
| 12     | $\mathrm{CO}_2$ column ratio $r_{\mathrm{CO}_2}$           | $0.99 < R_{\rm CO_2} < 1.018$                                                 |
| 13     | ${ m H_2O}$ column ratio $r_{{ m H_2O}}$                   | $0.95 < R_{\rm H_2O} < 1.08$                                                  |
| 14     | ${ m O_2}$ column ratio $r_{{ m O_2}}$                     | $0.96 < R_{\rm O_2} < 1.04$                                                   |

amount of good quality retrievals (Butz et al., 2010; Schepers et al., 2012). Such a set of criteria have been established also for GOSAT-2 and are listed in Table 1.

Criteria 6 to 9 are excluded for the Proxy product, since these are not in the state vector of the retrieval. Butz et al. (2010) defined the aerosol parameter as  $\omega = \tau_{\rm aer} \times 1/r_{\rm eff} \times z_{\rm aer}$  (Schepers et al., 2012). The blended albedo in criterion 10 is defined as  $A_{\rm bld} = 2.4 \cdot A(0.76\mu {\rm m}) - 1.13 \cdot A(2.0\mu {\rm m})$  with the retrieved albedos  $A(0.76\mu {\rm m})$  and  $A(2.0\mu {\rm m})$  at the indicated wavelengths (Wunch et al., 2011). Guerlet et al. (2013) investigated the use of the cirrus radiance for data filtering, which is defined as the mean radiance in the spectral range 5154.8–5157.8 cm<sup>-1</sup> (1.9388-1.9399  $\mu {\rm m}$ ). The use of the column ratios for data filtering was first proposed by Taylor et al. (2016) based on the difference in the non-scattering retrieved column from a weak and strong absorption band. For this purpose, in criteria 12 and 13, we use the CO<sub>2</sub> and H<sub>2</sub>O ratios inferred from the 1.6  $\mu {\rm m}$  and 2.0  $\mu {\rm m}$  spectral range. Finally, O<sub>2</sub> ratio is the retrieved O<sub>2</sub> column divided by the prior derived from the ECMWF surface pressure estimate.

## 4.2 Machine Learning Approach

An alternative approach to quality flagging with threshold criteria, as applied on GOSAT, is to use machine learning in the form of a random forest classifier (RFC). To this purpose, we use the RandomForestClassifier tool within Python's SciKit Learn package (Pedregosa et al., 2011).

A random forest model utilises an ensemble of N decision trees, which take a random subset of the available features and each make a decision on the target classification (Breiman, 2001). The final result of the model is taken as the majority chosen

class. This is ultimately applied to each ground pixel of GOSAT-2 data using a set of features consisting of RemoTeC retrieval parameters, to predict the quality of the retrieval. We use separately trained models for each of the three data products, which will be described in more detail in section 4.2.1.

# 4.2.1 RFC training using TCCON data

For the quality classification of our data product, we use a trained RFC. The supervised form of learning requires a labeled training dataset. To this end, we need knowledge of a "ground truth" and the best estimate of the true value of XCH<sub>4</sub> and XCO<sub>2</sub> comes from TCCON. Therefore, in order to determine the true label for the quality flag, we use GOSAT-2 level 2 data from measurements that are colocated in space and time with TCCON sites, and classify the training sample via the bias:

$$|\Delta X| < X_T$$
: label  $L_{X_T} = 0 \pmod{9}$ 

$$|\Delta X| > X_T$$
: label  $L_{X_T} = 1$  (bad) (10)

with the biases  $\Delta XCH_4 = XCH_{4,GOSAT-2} - XCH_{4,TCCON}$ , and  $\Delta XCO_2 = XCO_{2,GOSAT-2} - XCO_{2,TCCON}$ .  $X_T$  we name the training threshold and takes the form of e.g.  $\pm$  18 ppb for  $\Delta XCH_4$ . A label L of 0 corresponds to a good-quality retrieval, and a label of 1 means a bad-quality retrieval. For all training and validation, we use the TCCON GGG2020 release (Laughner et al., 2023).

A consequence of training the random forest model on GOSAT-2 colocations with TCCON is that retrievals with surface albedo  $\gtrsim 0.4$  were underrepresented in the training sample, due to the lack of TCCON stations in high albedo areas. This would lead to albedo-related biases when using such models to filter the global dataset. To avoid this, we defined a subsample of retrievals with albedo > 0.4 to include in the training set, using the threshold filtering criteria in Table 1.

Example ranges of albedo, along with several other geophysical parameters, covered in the combined training set are illustrated in Figure 2 for  $L_{X_T} = 0$ . A pre-flagging step is also applied which labels training data based on nonphysical values of albedo. The presence of retrievals with negative aerosol central height, or high optical depth (Fig 2), in the training set for high quality retrievals suggests that the training process may be improved by an stricter pre-flagging which includes aerosol related properties.

In this study, we limit the quality filtering of GOSAT-2 data using the machine learning approach to retrievals over land only, due to the lack of available training data over ocean. We note that this limiting factor would not apply to satellite data from push-broom spectrometers that have better spatial coverage, such as TROPOMI (Hu et al., 2016, 2018). Instead, we filter retrievals performed in glint mode over the ocean also following Table 1. Retrievals over ocean are discussed further in section A in the Appendix.

TCCON XCH<sub>4</sub> and XCO<sub>2</sub> data are also used to validate the final product (see section 5.1). Due to the supervised learning approach, utilising TCCON data in training the filtering models means that these data can no longer be quality filtered without receiving what was defined during training. Since the training data also comprise the validation data, this would lead to artificially choosing validation results, therefore compromising any independent validation with TCCON using the assigned quality flags. To avoid this, we train different filtering models, one year at a time, where the data from the year to be predicted are excluded from training. This results in one filtering model per year of data. Here we assume that the relationship between

retrieval quality and features is temporally independent. The robustness of this assumption is reflected in Figure 2 which shows a feature importance analysis of the different models for the XCH<sub>4</sub> Full Physics product. We see that there is, in general, little variation in the order of features over the different years, with the top four features always being the most important, showing that the models are all very similar.

## 4.2.2 RFC Prediction Performance

To evaluate our classification for the three products, we consider the performance of the RFCs by comparing its predicted labels to the true labels given by the elements of the confusion matrix (Liang, 2022): the False-Positive (FP), the False-Negative (FN), the True-Positive (TP), and True-Negative (TN). Here the terms 'true' and 'false' refer to a correct or wrong prediction, 'positive' and 'negative' to the bad and good label of the predicted data. From these, we evaluate the classification using the following metrics:

1. The true-positive-rate TPR is defined as

$$TPR = \frac{TP}{TP + FN}.$$
 (11)

and measures the number of correctly identified positive instances out of all true positive instances.

2. The False-Positive rate (FPR) is the corresponding rate of False Positive with respect to all true negative instances,

$$FPR = \frac{FP}{FP + TN} \,. \tag{12}$$

A binary classification model predicts the probability of an instance belonging to one of the two classes depending on the classification threshold, which we name  $p_t$ . Varying  $p_t$ , leads to the Receiver-Operating Characteristic curve (ROC) (Bradley, 1997), which is a parametric curve of  $FPR(p_t)$  versus  $TPR(p_t)$ . For a large threshold  $(p_t \to 1)$ , TPR goes to one, but so does the FPR. In the other extreme for  $p_t \to 0$ , both TPR and FPR go to zero. Therefore, the more the ROC curve goes through the top-left quadrant of the diagram, the better the classifier. This is characterized by the area under the ROC curve (AUC). We assume a value of 0.5 for  $p_t$  for all classification models.

Figure 2 compares the ROC curves of the XCH<sub>4</sub> Full Physics classification models to the XCH<sub>4</sub> Proxy ones. There is no clear differentiation between the ROC curves of each product, implying that the models for each year perform comparably to each other. Average metrics over all models per product are given in Table 2. From this we see that the performance of the RFC models for the two Full Physics products are similar - which is intuitive given that these come from the same retrieval - whereas the diagnostics for the Proxy product are slightly worse.

Such an effect can be understood by the nature of the Proxy approach and as a consequence of equation 5, where most of the systematic error is divided out by dividing the two columns of CH<sub>4</sub> and CO<sub>2</sub>. Consequently, the distinction between high quality and low quality retrievals is much more obvious in the Full Physics case. Quantitatively, the ratio of good to bad retrievals in the training data is about 0.3 for the Full Physics products, whereas for the Proxy it is 1.6. It is therefore easier to accurately label the training sample in the Full Physics RFC models, leading to better performance metrics.

**Figure 2.** (a) Histograms from the XCH<sub>4</sub> Full Physics filtering model for 2019, showing the ranges covered for various geophysical parameters in the training dataset, for good quality examples. Cirrus signal is given in units of mol. m<sup>-2</sup> s<sup>-1</sup> nm<sup>-1</sup> sr<sup>-1</sup>. Note that the albedo at 1629 nm is not used as a feature for training. (b) ROC curves for the different filtering models of the Full Physics XCH<sub>4</sub> product and Proxy XCH<sub>4</sub> product, in solid and dotted lines respectively. The solid grey line indicates the performance of a randomly guessed prediction, with a 50 % chance of being correct. (c) Feature importance of the XCH<sub>4</sub> Full Physics quality filtering models. The window numbers 1, 2, 3 and 4 correspond to the spectral bands 1, 2a, 2b and 3 of Figure 1, respectively. For the definitions of other features see section 4.1

# 4.2.3 The QA value

In the random forest models, the strictness of the training threshold,  $X_T$ , defined when labelling the training dataset (equations 9 & 10) has a directly proportional effect to the number of retrievals ultimately classed as good quality, as well as the scatter

**Table 2.** Summary of classification metrics averaged over all years for  $p_t$ =0.5.

| Product                       | TPR  | FPR  | AUC  |
|-------------------------------|------|------|------|
| XCO <sub>2</sub> Full Physics | 0.90 | 0.42 | 0.89 |
| XCH <sub>4</sub> Full Physics | 0.89 | 0.44 | 0.89 |
| XCH <sub>4</sub> Proxy        | 0.64 | 0.14 | 0.83 |

of the total column mixing ratio with respect to TCCON. Figure 3 shows the number of good retrievals as a function of the RMSE with TCCON derived for different training thresholds  $X_T$  and the depicted positive trend is intuitively expected. This allows us to define a non-binary quality flag that is grounded in TCCON validation.  $X_T$  is chosen to probe the steepest part of the curves in Figure 3, thus maximising the improvement that can be extracted from the machine learning filtering approach.

Starting with a set of n threshold values  $X_{T_1}, \dots X_{T_n}$  we can assign the label vector  $L = (L_1, \dots L_n)$  with  $L_i = L_{X_{T_i}}$  as defined in Eq. 9. We define the QA value of a data point by the mean value of the components of the corresponding label vector L,

$$QA = \langle L \rangle \tag{13}$$

QA can have n+1 discrete values in the range [0,1] depending on the number n of used threshold values  $X_T$ . For GOSAT-2, we use n=5 with  $QA \in [0,1/5,2/5,\cdots,1]$ .

For reference, Figure 3 also shows the results of filtering GOSAT-2 data using the thresholding defined in Table 1. For the Full Physics products, the new filtering can achieve an increase in data yield of  $\sim$  48 % and 85 % for XCH<sub>4</sub> and XCO<sub>2</sub> respectively, for equivalent RMSE. Alternatively, an improvement in RMSE of 2.2 ppb and 0.7 ppm for XCH<sub>4</sub> and XCO<sub>2</sub> respectively can be achieved for equivalent data yield. The larger improvement for XCH<sub>4</sub> compared to XCO<sub>2</sub> is a reflection of the less optimal choice of the arbitrary threshold criteria for XCO<sub>2</sub> (Table 1). For the Proxy product this can be 1.6 ppb for the same amount of data, or conversely, an increase of 29 % in data yield for the same RMSE. Thus user can therefore choose the option of more data, which may be advantageous to plume detection where better coverage is desirable, or better quality data, where as small as possible systematic biases are required by atmospheric modellers. Furthermore, with Figure 3, the user may choose the QA value which corresponds to their acceptable RMSE with TCCON.

# 5 Validation and Satellite Inter-comparison

## 5.1 TCCON Validation

TCCON is central to the work presented here as it provides both the ground truth in labeling training data, as well as one of the main validation sources. In this article, all references to TCCON are for the GGG2020 TCCON release (Laughner et al., 2023). The TCCON stations used in the analysis are summarised in the Appendix in Table B1.

**Figure 3.** Number of retrievals flagged as good for five different thresholds, as a function of the RMSE derived by the TCCON validation. The mean QA value per data ensemble is given by the color code. Results for XCH<sub>4</sub> are shown on the top panel, where the RFC filtering models for the Proxy and Full Physics product are represented by the dashed and solid lines respectively. On the bottom panel are the results for the XCO<sub>2</sub> filtering models. The red squares and triangles mark the parameter space for the statistics of filtering the data product according to Table 1.

In this section, we present the validation of our GOSAT-2 data products with respect to TCCON. TCCON sites are considered only if there are more than 50 spatio-temporal colocations with GOSAT-2 over the whole time-series, defined as overlying within a radius of 300 km and time range of  $\pm$  2.5 hrs. We evaluate the data products for the QA value of 0 (strictest filtering with RFC models; see section 4.2.3).

Figure 4 shows the correlation between colocated GOSAT-2 and TCCON data for XCO<sub>2</sub> and both XCH<sub>4</sub> products. These are single soundings of GOSAT-2 over land compared to an average of the TCCON measurements that coincide spatially and temporally. For XCH<sub>4</sub> we derive a RMSE of 15.2 ppb and 15.7 ppb for the Full Physics and Proxy products respectively, and Pearson's correlation coefficient of 0.89 and 0.88 for the Full Physics and Proxy products respectively. For XCO<sub>2</sub> these are 2.1 ppm and 0.88 respectively. For some stations, lines of data points in the x-axis direction are observed in Figure 4, which arise

from comparing daily averaged TCCON measurements to single soundings from GOSAT-2, indicative of bias with geolocation around a given site.

Time-series of GOSAT-2 colocations with TCCON for each product are shown in the Appendix section B. Following Noël et al. (2022), we further parameterise the bias over time as:

$$\Delta X = a_0 + a_1 t + a_2 \sin(2\pi t + a_3) + \epsilon \tag{14}$$

where equation 14 is fit to the time-series of the bias with each station individually.  $a_0$  is a constant bias term,  $a_1$  represents a linear term,  $a_2$  measures the amplitude of the seasonal variation of the bias,  $a_3$  measures the temporal shift of the seasonal term, and  $\epsilon$  is an error term.

The parameters in Table 3 are extracted from fits of equation 14 to the time-series of the bias for all TCCON stations. We illustrate an overview of the per station statistics in terms of site and seasonal bias, as well as linear drift in the bias, in Figures 5 to 7.  $\Delta_{site}$  is the site bias and defined as the mean of  $\Delta X$  from equation 14 and  $\Delta_{seas}$  is the seasonal bias and defined as the standard deviation of the seasonal (sine) term in equation 14. Finally,  $\Delta_{dri}$  is the linear drift and is calculated as  $a_1$  from equation 14.

For the Full Physics products, we derive average values of the site bias of -0.1 ppb and -0.2 ppm for XCH<sub>4</sub> and XCO<sub>2</sub> respectively, after bias correction. The seasonal bias term is higher for both products with 4.0 ppb and 0.6 ppm for XCH<sub>4</sub> and XCO<sub>2</sub> respectively. For the Proxy product, the average site bias and seasonal bias are 0.2 ppb and 3.1 ppb respectively. We exclude the station-averaged  $\Delta_{site}$  from Table 3 as it is by definition close to zero due to the bias correction. Before bias correction, the mean bias over all stations is 7.2 ppb and 13.3 ppb for the XCH<sub>4</sub> Full Physics and Proxy products, respectively. Thus, averagely speaking, the Full Physics retrieval approach is closer to the truth than the Proxy apporoach, before bias correction.

From Table 3 we also report a linear drift of 0.6 ppb yr<sup>-1</sup> and 0.2 ppm yr<sup>-1</sup> for XCH<sub>4</sub> and XCO<sub>2</sub> respectively, for the Full Physics products. For the Proxy product the average linear drift is 1.2 ppb yr<sup>-1</sup>. Another important metric of the systematic error is the station-to-station bias. This is defined as the standard deviation of the individual site biases, in contrast to the RMSE which is the standard deviation of all the differences together. We report station-to-station bias of 0.5 ppm, 4.2 ppb and 3.7 ppb for XCO<sub>2</sub>, XCH<sub>4</sub> Full Physics and XCH<sub>4</sub> Proxy, respectively. The site-to-site biases and linear drift terms are low, and below the breakthrough systematic error threshold requirements (GCOS, 2016), which is an essential characteristic of the data product for determining regional scale sources and sinks through flux inversion modelling.

We find that the difference between the average station RMSE and that calculated from the sample of GOSAT-2/TCCON differences as a whole can be significant. The RMSE for XCH<sub>4</sub> Full Physics taking all data as one sample is 15.2 ppb, however the average of the individual station RMSE is 13.1 ppb. From Figure 5, Caltech and Edwards have RMSE over 15 ppb, however the disproportionately high number of collocations (together constituting 40 % of the data points) skew significantly the statistics towards these stations. The location of these two stations in the Californian desert means more clear sky conditions, therefore a better coverage in the TCCON timeseries. Furthermore, the mid-latitude in the Northern hemisphere is favourable

**Figure 4.** GOSAT-2 XCO<sub>2</sub> (top), XCH<sub>4</sub> Full Physics (middle) and XCH<sub>4</sub> Proxy (bottom) plotted against TCCON, for retrievals over land. Data are compared only if they are fully colocated in space and time. The standard deviation of the population, Pearson's correlation coefficient and number of retrievals are given in the inset. The legend plots the different TCCON stations where markers are as follows. Stations that are along the coast and also sensitive to glint mode (ocean) measurements are indicated as circles. Those that have high latitudes in the northern and southern hemispheres are upward triangles and crosses, respectively. Stations in Asia, North America and Europe are indicated by squares, pluses and downward triangles respectively.

**Table 3.** Summary of the main statistics of GOSAT-2 product validation with TCCON. RMSE is the root mean square error,  $\Delta_{dri}$  is the linear drift and  $\Delta_{seas}$  is the seasonal bias, averaged over all stations.  $\sigma_{site}$  is the station-to-station bias.

| XCH <sub>4</sub> Full Physics |                 |                  | XCO <sub>2</sub> Full Physics |       |                 | XCH <sub>4</sub> Proxy |                 |       |                 |                  |                 |
|-------------------------------|-----------------|------------------|-------------------------------|-------|-----------------|------------------------|-----------------|-------|-----------------|------------------|-----------------|
| RMSE                          | $\Delta_{seas}$ | $\Delta_{dri}$   | $\sigma_{site}$               | RMSE  | $\Delta_{seas}$ | $\Delta_{dri}$         | $\sigma_{site}$ | RMSE  | $\Delta_{seas}$ | $\Delta_{dri}$   | $\sigma_{site}$ |
| (ppb)                         | (ppb)           | $(ppb\ yr^{-1})$ | (ppb)                         | (ppm) | (ppm)           | $(ppm \ yr^{-1})$      | (ppm)           | (ppb) | (ppb)           | $(ppb\ yr^{-1})$ | (ppb)           |
| 13.1                          | 4.0             | 0.6              | 4.2                           | 2.0   | 0.6             | 0.2                    | 0.5             | 14.7  | 3.1             | 1.2              | 3.7             |

to our GOSAT-2 products from RemoTeC. The combination of these two factors leads to much higher colocations than other stations, however they are know to be difficult for measuring GHG concentrations (Hedelius et al., 2017; Schneising et al., 2019). Taking this into consideration, we consider the values for XCH<sub>4</sub> in Figure 4 an upper limit. For the Proxy product the effect on RMSE is less although still notable, with 15.7 ppb compared to 14.7 ppb when taking the station-averaged RMSE.

Despite the lower performance of filtering models for the Proxy product compared to the Full Physics ones (section 4.2.2), the level 2 quality of the Proxy XCH<sub>4</sub> product presented in Table 3 is effectively as good as the Full Physics XCH<sub>4</sub> product, with the advantage of better data coverage. This can be understood by the ratio of FP/FN, for which in the case of Full Physics is 1:1, is 2:1 for the Proxy. The higher number of FPs lead to a poorer ROC curve, however in terms of the problem of quality filtering, FNs are more detrimental to the level 2 quality, since they correspond to ground truth bad data flagged as good.

For the operational GOSAT-2 products, Yoshida et al. (2023) report RMSE with respect to TCCON of 1.8 ppm and 8.9 ppb for XCO<sub>2</sub> and XCH<sub>4</sub> respectively, across a time range of March 2019 to Dec 2020. Also, they derive station-to-station bias of 0.71 ppm and 2 ppb for XCO<sub>2</sub> and XCH<sub>4</sub> respectively. We note the short time-series these values are derived from. Noël et al. (2021) find RMSE and station-to-station bias of 1.86 ppm and 1.14 ppm respectively, for XCO<sub>2</sub>. For XCH<sub>4</sub>, station-to-station biases of 4 to 6 ppb and RMSE of around 12 ppb are reported, for the and Full Physics and Proxy products Noël et al. (2022). The authors note that, due to the short time-series, these results are drawn from only seven TCCON stations, some of which span only a few months.

#### 5.2 GOSAT Inter-comparison

The similarity in the setup of GOSAT and GOSAT-2, along with the wide use of GOSAT in the scientific literature, make them ideal candidates for satellite inter-comparison. We compare our GOSAT-2 Full Physics products with the corresponding GOSAT products from RemoTeC, version 2.3.8, over time frame of 2019 to 2023. For the Proxy product comparison, we compare our GOSAT-2 XCH<sub>4</sub> Proxy product to that of GOSAT version 2.3.9.

The data from the two satellites are matched by re-gridding XCH<sub>4</sub> to  $2^{\circ} \times 2^{\circ}$  lat/lon boxes, per day. A colocation is considered when there are data from each satellite in the same grid cell for a given day. GOSAT data are quality filtered using the filters listed in Table 1 with slightly different values, thus RFC filtering is applied only to GOSAT-2. We present comparisons only for the GOSAT-2 QA value of 0.

Figure 5. Overview of the bias parametrisation for the Full Physics XCH<sub>4</sub> product, per station. Shown in blue is the RMSE, red the site bias  $\Delta_{site}$ , green the linear drift  $\Delta_{dri}$ , yellow the seasonal bias  $\Delta_{seas}$  and in purple the number of retrievals. Values for GOSAT-2 are shown in bold bars, and those of GOSAT are in light bars. Stations are listed in order of decreasing latitude. Missing bars correspond to less than 50 colocations for that station, therefore we do not calculate the values there. We note that the site bias for GOSAT-2 at Bremen is close to zero.

Figure 6. Overview of the bias parametrisation for the Full Physics  $XCO_2$  product, per station. Shown in blue is the RMSE, red the site bias  $\Delta_{site}$ , green the linear drift  $\Delta_{dri}$ , yellow the seasonal bias  $\Delta_{seas}$  and in purple the number of retrievals. Values for GOSAT-2 are shown in bold bars, and those of GOSAT are in light bars. Stations are listed in order of decreasing latitude. Missing bars correspond to less than 50 colocations for that station, therefore we do not calculate the values there.

Figure 7. Overview of the bias parametrisation for the Proxy XCH<sub>4</sub> product, per station. Shown in blue is the RMSE, red the site bias  $\Delta_{site}$ , green the linear drift  $\Delta_{dri}$ , yellow the seasonal bias  $\Delta_{seas}$  and in purple the number of retrievals. Values for GOSAT-2 are shown in bold bars, and those of GOSAT are in light bars. Stations are listed in order of decreasing latitude. Missing bars correspond to less than 50 colocations for that station, therefore we do not calculate the values there.

**Figure 8.** GOSAT-GOSAT-2 comparison for the GOSAT-2 XCH<sub>4</sub> Full Physics product. Maps are shown of XCH<sub>4</sub> over the year 2020 averaged onto  $2^{\circ} \times 2^{\circ}$  boxes for GOSAT and GOSAT-2 on the left and right respectively.

From the global maps of the XCH<sub>4</sub> Full Physics product in Figure 8, the superior coverage of GOSAT-2 is striking; a consequence of the intelligent pointing system to avoid cloudy scenes. Maps for the other two products are shown in Figure C1 in the Appendix. We further analyse how the GHG concentrations compare, illustrated as kernel density estimation (KDE) plots, analysing data over land only. The scatter of satellite differences is 14.5 ppb, similar to the RMSE of the bias with TCCON (13.1 ppb; see section 5.1). We find a large average global bias of -15.2 ppb, which we discuss further in section 5.3.

For the Proxy XCH<sub>4</sub> product, the comparison between GOSAT and GOSAT-2 is better than the Full Physics product. The average global bias is only -5.3 ppb, and the standard deviation and correlation coefficients are 13.5 ppb and 0.9 respectively.

For XCO<sub>2</sub>, the correlation between GOSAT and GOSAT-2 is weaker, with a coefficient of 0.64 compared to 0.88 for XCH<sub>4</sub>. This difference is expected, as CO<sub>2</sub>'s longer atmospheric lifetime leads to greater large-scale diffusion, reducing correlation strength. The scatter of the differences is 2.9 ppm, slightly higher than the GOSAT-2 RMSE with respect to TCCON of 2.0 ppm, and we find a bias of 0.9 ppm.

Furthermore, we plot time-series of GOSAT and GOSAT-2 globally, and for the three latitude bands of Northern/Southern Hemispheres (NH & SH) and the Tropics in Figure 10. These are defined as  $0^{\circ}$  to  $60^{\circ}$ N for NH,  $-25^{\circ}$ N to  $25^{\circ}$ N for the Tropics, and  $-60^{\circ}$ N to  $0^{\circ}$  for SH.

The globally averaged seasonal cycles of XCH<sub>4</sub> Full Physics follow each other well between April and August, but from September to March, the GOSAT one peaks at higher values. This characteristic is representative of the Tropics and SH timeseries, however for the NH time-series, the GOSAT time-series is consistently higher by approximately 15 ppb.

For the time-series of the Proxy products, we find that the satellite time-series correlate well with each other. The seasonal cycles follow each other closely in all latitude bands, however the bias begins positive but then switches around the halfway point of the time-series.

For XCO<sub>2</sub>, the time-series in the NH follow each other closely until mid-2020 after which the GOSAT time-series in consistently higher than GOSAT-2. The SH time-series agree well over the whole time-series, but that of the Tropics is less pronounced in GOSAT-2 with larger seasonal fluctuations exhibited for GOSAT.

**Figure 9.** Correlation between GOSAT and GOSAT-2 shown as a kernel density estimation (KDE) plots for each data product. Plots for XCH<sub>4</sub> Full Physics, XCO<sub>2</sub> Full Physics and XCH<sub>4</sub> Proxy are shown from left to right, respectively. The mean bias, standard deviation, number of points and correlation coefficient of the population are also quoted. Histograms of the number of counts are shown around the margin, along with the linear regression and the 1-to-1 lines in black and grey respectively. Results are for soundings over land.

Comparing the TCCON validation for the GOSAT-2 data products to those of GOSAT (Figs 5 to 7), we find that generally the RMSE is lower for GOSAT-2 than GOSAT across all stations, while the number of retrievals is higher for GOSAT-2. We observe that the site bias is smaller for GOSAT-2, with GOSAT showing some significant biases with respect to TCCON, whereas the linear drift is more variable between the two satellites. We note here that we compare data products from GOSAT and GOSAT-2. We do not comment on the performance of one satellite over another as the data products use different quality filtering methods. A more concrete comparison could be made by applying RFC quality filtering to GOSAT, however this is out of the scope of this paper.

### 5.3 TROPOMI Intercomparision

The fact that the RFC quality filtering models are trained on the spatially limited dataset of TCCON implies that understanding how well the models - and thus also the filtering - generalise to the global GOSAT-2 dataset, is of high priority. This is reinforced when considering that the validation data and the training data constitute essentially the same representation of data, which may lead to biases that would not be picked up by validation with TCCON only. Central to the performance of the models is the behaviour exhibited in Figure 3. Therefore if such behaviour is exhibited also on global scales, this is good confirmation that the quality filtering performs equivalently on the global dataset as it does on data colocated with TCCON.

Here we inter-compare our GOSAT-2 product against the TROPOMI operational product, version 2.4.0, and evaluate the performance of the quality filtering on global scales. The TROPOMI product was pre-filtered with VIIRS cloud product using the strictest filter of cloud fraction < 0.001, and quality filtered using nominal quality flags. The same colocation criteria are used as for the GOSAT inter-comparison. We note that since no XCO<sub>2</sub> product exists for TROPOMI, the inter-comparison here is limited to the XCH<sub>4</sub> products.

**Figure 10.** Time-series of the GOSAT and GOSAT-2 RemoTeC products. The XCH<sub>4</sub> Full Physics, XCO<sub>2</sub> Full Physics and XCH<sub>4</sub> Proxy products are shown from left to right. GOSAT-2 data are shown as solid lines, whereas GOSAT data are shown as dashed lines. The upper panels give the globally averaged monthly time-series. The lower panels give the same but split into the different latitude bands of NH, SH and Tropics. For XCH<sub>4</sub>, the time-series of the Tropics are shifted up by a constant factor of +50 ppb for better visualisation. For XCO<sub>2</sub>, the time-series of the Tropics is shifted by +20 ppm and the SH by -10 ppm.

Figures 11 and 12 illustrate results for the whole of the year 2020, taking GOSAT-2 QA value equal to 0, for the Full Physics and Proxy XCH<sub>4</sub> products respectively. To evaluate the generalisation of the RFC quality filtering to global scales, we give results for the other QA values of the GOSAT-2 product in Table 4. Furthermore, because the RFC filtering in GOSAT-2 is only applied to soundings over land, we restrict the analysis to satellite data over land.

We find that, when considering GOSAT-2 data with QA value of 0, the global systematic bias between GOSAT-2 and TROPOMI, which we define as XCH<sub>4,GOSAT-2</sub>-XCH<sub>4,TROPOMI</sub>, is very low. We derive a global average of the bias of -4.6 ppb and 1.7 ppb for the Full Physics and Proxy products respectively. The satellite products are highly correlated with correlation coefficients above 0.88 and 0.87, and standard deviations of 15.0 and 16.6 ppb of XCH<sub>4</sub> Full-Physics and Proxy data. Here we note that the TROPOMI operational product uses a different bias correction to GOSAT-2. The TROPOMI bias correction is based on the small area approximation (Lorente et al., 2021; O'Dell et al., 2018) taking a uniform XCH<sub>4</sub> distribution as a function of albedo in multiple regions, whereas the GOSAT-2 bias correction is based on TCCON data (equation 8).

A key result shown in Table 4 is that the QA value increases proportional to the scatter of the GOSAT2-TROPOMI differences and number of data points. This is a good reflection of the behaviour represented in Figure 3, meaning that, despite the fact that the quality filtering models are trained on the spatially limited dataset of TCCON, they generalise well to the global ensemble. For reference, the statistics of the TCCON validation of every GOSAT-2/TCCON colocation are also given in Table 4, for each QA value.  $\sigma_{\text{TCCON}}$  is calculated as the average RMSE over all stations. The bias for the GOSAT-2 Full Physics product systematically increases with QA value. That of the Proxy product looks to decrease, however the change of 0.7 ppb can be treated as negligible.

From the global maps, significant differences between TROPOMI and GOSAT-2 are obvious over Northern/Central Africa, and we speculate that these differences may be attributed to dust and burning events that lead to high aerosol load, thus making

the retrieval more difficult. This conclusion would be consistent with the fact that the biases are larger for the Proxy GOSAT-2 product than the Full Physics product, in which aerosols are better characterised. The reason for low coverage and high bias over the Amazon can be a result of low surface albedo and observations that are contaminated by high water vapour.

To ascertain whether the different labelling of training data in the RFC filtering models introduces a differently biased data product depending on the albedo, we also looked at soundings with albedo greater than 0.4 only. Here we exclude North Africa due to the large apparent biases. We find average differences of -2.6 ppb and -2.1 ppb for the Full Physics and Proxy and products, respectively. These results are very similar to those in Table 4, implying that the quality filtering is consistent throughout the entire albedo range.

The aggregate global difference between TROPOMI and GOSAT-2 is close to zero, in contrast to what we observe for GOSAT. Systematic biases of -13 ppb are found between TROPOMI and GOSAT for a global average (Hu et al., 2018; Lorente et al., 2021). A similar bias is found, in both sign and magnitude, between GOSAT and GOSAT-2 (section 5.2). We propose therefore that the bias observed between GOSAT and GOSAT-2 comes from systematic biases in the GOSAT XCH<sub>4</sub> products, consistent with the results of TCCON validation presented in Figures 5 and 7.

**Table 4.** Overview of inter-comparison of XCH<sub>4</sub> between GOSAT-2 and TROPOMI. Information for all QA values available to GOSAT-2 are given.  $\Delta$ XCH<sub>4</sub> is the mean bias with TROPOMI,  $\sigma_{\text{TROPOMI}}$  and  $N_{\text{TROPOMI}}$  are the scatter and number of TROPOMI-GOSAT-2 colocations respectively, and  $N_{\text{TCCON}}$  is the number of TCCON-GOSAT-2 colocations, with  $\sigma_{\text{TCCON}}$  the RMSE of the bias between GOSAT-2 and TCCON.

| XCH <sub>4</sub> Full Physics |                      |                            |                        |                |                            |
|-------------------------------|----------------------|----------------------------|------------------------|----------------|----------------------------|
| QA value                      | $\Delta XCH_4$ (ppb) | σ <sub>TROPOMI</sub> (ppb) | $N_{\mathrm{TROPOMI}}$ | $N_{ m TCCON}$ | $\sigma_{\rm TCCON}$ (ppb) |
| 0                             | -4.6                 | 15.0                       | 22,863                 | 17,250         | 13.1                       |
| 0.2                           | -4.8                 | 15.4                       | 29,539                 | 22,635         | 13.7                       |
| 0.4                           | -5.3                 | 16.5                       | 38,049                 | 28,943         | 15.3                       |
| 0.6                           | -5.7                 | 17.4                       | 43,059                 | 32,309         | 16.6                       |
| 0.8                           | -6.3                 | 18.5                       | 47,896                 | 34,578         | 17.7                       |
| XCH <sub>4</sub> Proxy        |                      |                            |                        |                |                            |
| QA value                      | $\Delta XCH_4 (ppb)$ | σ <sub>TROPOMI</sub> (ppb) | $N_{\mathrm{TROPOMI}}$ | $N_{ m TCCON}$ | $\sigma_{\rm TCCON}$ (ppb) |
| 0                             | 1.7                  | 16.6                       | 76,353                 | 55,915         | 14.7                       |
| 0.2                           | 1.8                  | 17.3                       | 77,540                 | 63,248         | 15.3                       |
| 0.4                           | 1.5                  | 18.2                       | 88,983                 | 67,607         | 15.8                       |
| 0.6                           | 1.2                  | 19.0                       | 93,385                 | 70,198         | 16.1                       |
| 0.8                           | 1.0                  | 19.7                       | 97,451                 | 71,884         | 16.4                       |
| -                             |                      |                            |                        |                |                            |

Figure 11. GOSAT2-TROPOMI comparison with QA value equal to 0 for the GOSAT-2 Full Physics product.  $upper\ left$ : Map of TROPOMI XCH<sub>4</sub> daily averages colocated with GOSAT-2, over the year 2020, sampled on  $2^{\circ} \times 2^{\circ}$  boxes.  $upper\ right$ : Map of GOSAT-2 XCH<sub>4</sub> daily averages colocated with TROPOMI, over the year 2020, sampled on  $2^{\circ} \times 2^{\circ}$  boxes.  $lower\ left$ : Map of the difference between satellite data defined as XCH<sub>4,GOSAT-2</sub>-XCH<sub>4,TROPOMI</sub>.  $lower\ right$ : Correlation between all colocated XCH<sub>4</sub> measurements over 2020, shown as a kernel density estimation (KDE) plot. The mean bias, standard deviation, number of points and correlation coefficient of the population are also quoted. Histograms of the number of counts are shown around the margins, along with the linear regression and the 1-to-1 lines in black and grey respectively.

## 6 Conclusions

In this article, we have presented total column mixing ratio data products from GOSAT-2, retrieved with the RemoTeC algorithm. From the two retrieval approaches of RemoTeC, three products are extracted; XCH<sub>4</sub> and XCO<sub>2</sub> from the Full Physics retrieval and XCH<sub>4</sub> from the Proxy retrieval. The time-series of these products span five years, from 2019 to 2023. All three products are validated with TCCON and inter-compared to GOSAT and TROPOMI and the long time-series ensures robust results from each.

Figure 12. The same as Figure 11 but for the GOSAT-2 Proxy product.

The RMSE between GOSAT-2 and TCCON of both the XCH<sub>4</sub> products is below 15 ppb, with the Proxy product having more data by a factor of 3, and the RMSE of XCO<sub>2</sub> is 2 ppm. We derive station-to-station biases of 4.2 ppb and 0.5 ppm for the XCH<sub>4</sub> and XCO<sub>2</sub> Full Physics products respectively, and 3.7 ppb for the Proxy product. Finally we quantify the linear drift as 0.6 ppb yr<sup>-1</sup>, 0.2 ppm yr<sup>-1</sup> and 1.2 ppb yr<sup>-1</sup> for the XCH<sub>4</sub> Full Physics, XCO<sub>2</sub> Full Physics and XCH<sub>4</sub> Proxy products respectively.

In comparison to GOSAT, the GOSAT-2 XCH<sub>4</sub> Full Physics product shows large differences, with a global average bias of -15 ppb. This is less so for the Proxy product and on the order of -5 ppb. Compared to TROPOMI, GOSAT-2 is in excellent agreement, with average global biases of -4.6 ppb and 1.7 ppb for the Full Physics and Proxy GOSAT-2 products respectively. High correlation coefficients above 0.85, and standard deviations less than 17 ppb are derived for GOSAT-2 compared to TROPOMI.

Finally, we present a new quality filtering based on a machine learning approach. Training data for the random forest classifier models are taken from TCCON colocations with GOSAT-2, where we classify good/bad quality retrievals through

the bias with TCCON. Since TCCON data are also used to validate the products, we train separate models to quality filter each year of data, to avoid compromising any independent validation.

Multiple QA values are implemented by training models with different training thresholds. Increasing the QA value leads to more data at the cost of worsening the RMSE with TCCON. In this way, users can choose between higher data yield or better quality data, which may have different advantages depending on the use of the data product.

## 465 Appendix A: GOSAT-2 data over Ocean

Despite the low surface albedo, satellite measurements over ocean are possible when operating the satellite in sunglint mode. Sunglint observations take advantage of specific viewing angles where the radiance of back-scattered sunlight is higher due to reflection from waves. This amplifies the albedo, allowing retrievals over ocean to be carried out, where the albedo is generally too low to retrieve accurate concentrations.

Figure A1 shows XCH<sub>4</sub> single soundings over land and ocean spatially averaged in latitude/longitude to  $2^{\circ} \times 2^{\circ}$ . Data are for 6 consecutive days, which is the GOSAT-2 revisit time, thus no temporal averaging occurs. We apply a different bias correction to retrievals over ocean, although it is very similar to the correction for land retrievals (sec 3.4). Again we use a simple empirical relation:

$$X_{corr} = X_{ret}(a + b\theta) \tag{A1}$$

where  $X_{ret}$  and  $X_{corr}$  are the bias corrected and retrieved concentrations respectively,  $\theta$  is the ratio of the retrieved  $O_2$  column to the prior, and a and b are determined such that the difference with TCCON is minimised. Visually, from Figure A1, there are no obvious differences between land and ocean with the latitudinal gradient captured in both.

TCCON stations are located only on land, therefore validation of sunglint observations are only possible using stations that are close to shorelines, or on islands. In this section, the results of the TCCON validation for sunglint mode, for all three RemoTeC GOSAT-2 products, are presented and shown in Figures A2 to A5.

Figure A1. Map of XCH<sub>4</sub> from the Proxy retrieval for 6 consecutive days (GOSAT-2 revisit time) in Spring on a  $2^{\circ} \times 2^{\circ}$  coordinate grid.

**Table A1.** TCCON validation of GOSAT-2 data products for stations with measurements over both land and ocean. Stations marked with an asterisk have fewer than 5 colocations in glint mode, so should be treated with caution.

| XCH <sub>4</sub> Full Physics |                                       |      | $XCH_4$          | Proxy           | XCO <sub>2</sub> Full Physics |                 |  |
|-------------------------------|---------------------------------------|------|------------------|-----------------|-------------------------------|-----------------|--|
| Station                       | tion Ocean Bias (ppb) Land Bias (ppb) |      | Ocean Bias (ppb) | Land Bias (ppb) | Ocean Bias (ppm)              | Land Bias (ppm) |  |
| Burgos                        | -2.4                                  | 0.1  | -5.7             | 3.1             | 1.0                           | -0.3            |  |
| Darwin                        | -10.1                                 | -7.6 | -3.3             | -1.8            | -1.5                          | -1.3            |  |
| Lauder                        | -0.8                                  | 1.4  | -4.6             | 3.5             | -0.1                          | 0.3             |  |
| Rikubetsu*                    | -7.1                                  | 3.8  | -10.6            | 9.8             | 3.3                           | 0.1             |  |
| Saga*                         | -6.9                                  | 0.4  | -1.8             | 3.5             | 2.9                           | 0.4             |  |
| Wollongong                    | -4.1                                  | -3.1 | -10.7            | 1.7             | 0.9                           | -0.3            |  |

The RMSE for ocean measurements is higher than over land, although correlation coefficients are comparable. For XCH<sub>4</sub>, this is more obvious for the Full Physics product, compared to the Proxy product, with 3 ppb difference in RMSE between ocean and land. We note that such statistics are drawn only from a handful of TCCON stations due to the limited availability of TCCON data close to the ocean. As mentioned in section 4.2.1, GOSAT-2 measurements in sunglint mode are quality filtered using the threshold criteria described in section 4.1.

In Table A1 we show the bias per product for TCCON stations that have measurements over both land and ocean. We note that, due to the limited number of colocations in glint mode, we calculate bias as the median of all colocations per station, unlike Figures 5 - 7 which fit equation 14 to the bias timeseries. The Full Physics XCH<sub>4</sub> product shows the best agreement between land and ocean with maximum differences of around 3 ppb, excluding Rikubetsu and Saga which have 1 data point each over ocean. The XCO<sub>2</sub> Full Physics product also has good agreement between land and ocean with differences of at most 0.35 % of CO<sub>2</sub>. The Proxy XCH<sub>4</sub> product however shows large differences between land and ocean, with even the sign of the bias changing for most stations, and differences on average to 0.5 % of CH<sub>4</sub>, pointing to land/ocean biases potentially caused by the different quality filtering applied over land and ocean.

### **Appendix B: Supplementary Material of TCCON Validation**

Here we provide additional information on the validation of GOSAT-2 products with TCCON. Table B1 lists all the TCCON stations used in the analysis. Data from all stations are also used as input to train the RFC quality filtering networks. Figures B1 to B3 present time-series of GOSAT-2 compared to TCCON for all stations for the XCH<sub>4</sub> Full Physics, XCO<sub>2</sub> Full Physics and XCH<sub>4</sub> Proxy products respectively. When enough TCCON data is available, time-series span the full 5 year period from 2019 to 2023.

**Figure A2.** GOSAT-2 plotted against TCCON for the Full Physics XCH<sub>4</sub>, Full Physics XCO<sub>2</sub> and Proxy XCH<sub>4</sub> products from left to right respectively. Data are compared only if they are fully colocated in space and time. The standard deviation of the population, Pearson's correlation coefficient and number of retrievals are given in the inset. The legend plots the different TCCON stations.

**Figure A3.** Time-series of GOSAT-2 colocated measurements over ocean with TCCON stations for the XCH<sub>4</sub> Full Physics retrievals. All GOSAT-2 observations are taken in sunglint mode. Pink circles correspond to the daily average of TCCON soundings that are spatio-temporally colocated with GOSAT-2. All individual GOSAT-2 sounding coloated with TCCON are plotted as blue circles, and the daily average of these are given as black triangles.

Figure A4. Same as Figure A3 but for the XCO<sub>2</sub> Full Physics product.

**Figure A5.** Same as Figure A3 but for the XCH<sub>4</sub> Proxy product.

**Table B1.** List of TCCON stations used in training the quality filtering model and/or validation.

| Site (Country)           | Coordinates (lat, lon°) | Temporal Extent     | Reference                     |
|--------------------------|-------------------------|---------------------|-------------------------------|
| Bremen (Germany)         | [53, 8.85]              | Jan 2009 - May 2021 | (Notholt et al., 2022)        |
| Burgos (Phillipines)     | [18.53, 120.65]         | Feb 2017 - Nov 2022 | (Morino et al., 2022c)        |
| Caltech (USA)            | [34.14, -118.13]        | Aug 2012 - Aug 2023 | (Wennberg et al., 2022a)      |
| Darwin (Australia)       | [-12.42, 130.89]        | Jan 2013 - Dec 2022 | (Deutscher et al., 2023b)     |
| East Trout Lake (Canada) | [54.35, -104.99]        | Sep 2016 - Sep 2023 | (Wunch et al., 2022)          |
| Edwards (USA)            | [34.96, -117.88]        | Jun 2013 - Aug 2023 | (Iraci et al., 2022)          |
| Eureka (Canada)          | [80.05, -86.42]         | Jun 2010 - Jun 2020 | (Strong et al., 2022)         |
| Garmisch (Germany)       | [47.48, 11.06]          | Jun 2007 - Apr 2023 | (Sussmann and Rettinger, 2023 |
| Harwell (UK)             | [51.57, -1.32]          | Apr 2021 - Aug 2023 | (Weidmann et al., 2023)       |
| Hefei (China)            | [31.9, 119.17]          | Oct 2015 - Nov 2022 | (Liu et al., 2023)            |
| Izana (Spain)            | [28.30, 16.50]          | Jan 2014 - Jul 2023 | (García et al., 2022)         |
| Karlsruhe (Germany)      | [49.10, 8.44]           | Jan 2014 - May 2023 | (Hase et al., 2023)           |
| Lamont (USA)             | [36.60, -97.49]         | Mar 2011 - Jul 2023 | (Wennberg et al., 2022c)      |
| Lauder (New Zealand)     | [-45.04, 169.68]        | Sep 2018 - Feb 2023 | (Pollard et al., 2022)        |
| Nicosia (Cyprus)         | [35.14, 33.38]          | Aug 2019 - May 2021 | (Petri et al., 2024)          |
| Ny Ålesund (Norway)      | [78.92,11.92]           | Feb 2005 - Aug 2022 | (Buschmann et al., 2022)      |
| Orleans (France)         | [47.97, 2.11]           | Aug 2009 - Nov 2022 | (Warneke et al., 2022)        |
| Paris (France)           | [48.49, 2.36]           | Aug 2014 - May 2023 | (Té et al., 2022)             |
| Park Falls (USA)         | [45.95, -90.27]         | May 2004 - Jul 2023 | (Wennberg et al., 2022b)      |
| Reunion Island (France)  | [-20.9, 55.48]          | Feb 2015 - Jun 2020 | (De Mazière et al., 2022)     |
| Rikubetsu (Japan)        | [43.46, 143.77]         | May 2014 - May 2021 | (Morino et al., 2022a)        |
| Saga (Japan)             | [33.24, 130.29]         | Jun 2011 - Sep 2022 | (Shiomi et al., 2022)         |
| Sodankylå (Finland)      | [67.37, 26.63]          | Apr 2009 - Apr 2023 | (Kivi et al., 2022)           |
| Tsukuba (Japan)          | [36.05, 140.12]         | Feb 2014 - Feb 2021 | (Morino et al., 2022b)        |
| Wollongong (Australia)   | [-34.41, 150.88]        | Jan 2013 - Feb 2023 | (Deutscher et al., 2023a)     |
| Xianghe (China)          | [39.80, 116.96]         | May 2018 - Apr 2022 | (Zhou et al., 2022)           |

**Figure B1.** Time-series of GOSAT-2 colocated measurements with TCCON stations for the XCH<sub>4</sub> Full Physics retrievals. Pink squares correspond to the daily average of TCCON soundings that are spatio-temporally colocated with GOSAT-2. All individual GOSAT-2 sounding coloated with TCCON are plotted as blue circles, and the daily average of these are given as black triangles.

Figure B2. Same as Figure B1 but for the XCO<sub>2</sub> Full Physics product.

Figure B3. Same as Figure B1 but for the XCH<sub>4</sub> Proxy product.

## 500 Appendix C: Supplementary Material GOSAT Intercomparison

Figure C1 shows the comparison between global maps of GOSAT and GOSAT-2, for the XCH<sub>4</sub> Proxy and XCO<sub>2</sub> products, highlighting the much improved spatial coverage of the GOSAT-2 products. Visually, the distribution of XCO<sub>2</sub> and XCH<sub>4</sub> are very similar between GOSAT and GOSAT-2, with hot-spots, at least for XCH<sub>4</sub>, in all the same places.

Figure C1. GOSAT-GOSAT-2 comparison for the GOSAT-2  $XCO_2$  Full Physics (top) and  $XCH_4$  Proxy (bottom) products. Maps are shown over the year 2020 averaged onto  $2^{\circ} \times 2^{\circ}$  boxes for GOSAT and GOSAT-2 on the left and right respectively.

Author contributions. AB ran the retrievals to obtain level 2 data from GOSAT-2, conducted the analysis on all three products and wrote the paper. MM provided the TROPOMI operational product data. TB and JL gave guidance in writing the paper and significant input into the configuration and architecture of the retrieval. All other co-authors provided data from the TCCON network.

*Competing interests.* Three of the co-authors are members of the editorial board for Atmospheric Measurement Techniques in the subject area of Gases.

Acknowledgements. The research in this manuscript was supported by funding from the European Space Agency (ESA) via the project 510 GHG-CCI+ (ESA contract no. 4000126450/19/I-NB). We thank the European Center for Medium Range Weather Forecasts (ECMWF) for providing the ERA5 reanalysis data. We also thank ESA and JAXA/NIES for providing us with the GOSAT-2 level 1b data. MV acknowledges Horizon 2020 (grant no. EMME-CARE – Eastern Mediterranean and Middle East – Climate and Atmosphere Research Centre (856612)).

Data availability. All three GOSAT-2 products are operationally provided as part of the Copernicus Climate Change Service (C3S) and can be downloaded from https://zenodo.org/records/12180512 under version 2.1.0 (DOI:10.5281/zenodo.12180512). Alternatively all three products will be made available on the CEDA Data Archive, https://catalogue.ceda.ac.uk/, as part of Climate Change Initiative plus (CCI+) under version v2.0.3.

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
