# Peer review of "Years of GOSAT-2 Retrievals with RemoTeC: XCO2 and XCH4 Data Products with Quality Filtering by Machine Learning"

_EGUsphere, 2024_

## Author Response (AR1)

**To the editor:**

We thank the referees for their kind words and constructive feedback. To improve the manuscript accordingly we will incorporate each of the points raised in the revised paper. We give responses to each comment below. Comments from the referees are given followed by our response. We give line numbers and Table/Figure numbers according to the revised manuscript. The response to the first reviewer is followed by the response to the second. At the end you will find a list of the most notable revisions.

**Response to RC1 – Rob Parker**

L12 - Do TCCON/GOSAT-2 co-locations sample an adequate range of geophysical parameters (aerosols, albedos, etc) to produce a robust post-filter? Many approaches use multiple "truth metrics" (e.g. TCCON, models, small area approximation, etc). You deliberately include some high-albedo cases into your training data to compensate for this but do I understand correctly that you do not have TCCON co-locations for these? How confident are you that this data is not biased differently?

The referee raises here an important point about the data used for training the machine learning filtering models. There are two questions asked here: 1) Is an adequate parameter space covered during training of the models? and 2) Does the different labelling of training data, depending on the albedo, lead to different results in the final product? We answer these both one by one.

i)Figure 2 has been expanded to present the ranges of a number of variables - including albedo, solar zenith angle, surface elevation variance among others - for high quality examples only. Here we choose the 2019 filtering model for the XCH4 Full Physics product. The combined training dataset sufficiently covers the expected ranges of these geophysical parameters. We discuss these additional panels in the text at Lines 226 – 230.

ii)Due to the coverage of stations, training data from TCCON alone do not cover the full albedo range in the GOSAT-2 data. To avoid potentially biasing the filtering models to lower albedo, training data were supplemented with a differently defined sample of data that includes soundings where the surface albedo is higher than 0.4. The additional training subset cannot be labelled in the same way as the TCCON subset, therefore we label it according to Table 1, and also cannot be validated in the same way either. To make sure there is not a significant different between the filtered data products based on the albedo, we have included an additional analysis between GOSAT-2 and TROPOMI, for albedo >0.4 only, and discuss this at Line 431. For estimating the bias, we exclude North Africa from the comparison due to the large bias in this region apparent in Figures 11 and 12. We find that the average bias of this sub-population is 2.1 ppb for the

Proxy product and -2.6 ppb for the Full Physics product, which are in close agreement with that from Table 4. The manuscript has been updated accordingly.

L24 - Why do GOSAT and GOSAT-2 disagree (with the same algorithm and priors?) compared to TROPOMI?

The literature does not provide a clear conclusion on this point. In this paper we consider the longest timeseries in the literature, however the magnitude of the difference between GOSAT and GOSAT-2 level 2 products is partially shrouded by the different quality filtering methods applied (threshold filtering for GOSAT and RFC filtering for GOSAT-2). It could be fortuitous that GOSAT-2 and TROPOMI agree equally bad compared to GOSAT, however given the improved quality filtering of GOSAT-2 we state in the text (L439) that the disagreement between GOSAT and GOSAT-2 is due to biases in the GOSAT products.

L31 - Some minor typos should be corrected, e.g. "anomoly".

Typo has been corrected.

L72 - Is the stated TCCON performance (L72) relevant for GGG2020 or does it relate to older GGG2024 data?

The performance stated is for the GGG2014 release. This has been clarified in the text (L80).

L83 - Rather than footnotes for the ESA CCI documents (L83/84), can they please be cited fully in the bibliography?

Full citations added for both PUGs and ATBDs for the Proxy and Full Physics products.

L87 - The instrument line shape for GOSAT-2 potentially has caused some issues with other retrieval algorithms. Could you please elaborate on its usage here (L87) and whether, if at all, you do anything to compensate for these (e.g. fitting a shift/stretch). I would also **strongly** recommend including a figure of a typical spectral residual for each species so the quality of the fit can be shown.

No stretch/squeeze or shift quantities are retrieved or fit to the ILS. We simply use it as is. A statement on this has been added to the text (L154). Following the referee's recommendation, we have included an example of spectral fits per band for the Physics retrieval in Figure 1, which shows the measurement, model and residuals for a high quality retrieval. L119 cites this figure.

L147 – Would it be possible to outline your full state vector (or explicitly link back to section/table in previous work where this is fully described)?

The state vectors are fully described in the CCI+ ATBDs. We have given the citation and exact location of where it can be found at L154 and L159.

Table 1 – Can it please be made explicit in the table which of these do not apply to the Proxy (i.e. 6-8?).

We have now marked filters that do not apply to the Proxy product with an asterisk and also clarify this in the caption of Table 1.

L157 - Can you specify which TCCON version is used for this study (and ideally the temporal extent of the different TCCON datasets)?

Clarification that we use the GGG2020 release added at L170. The temporal extent at each site has been added to Table B1.

L158 – Given that the XCH4 FP and Proxy (post-bias correction) have quite different biases, can you also compare the non-bias corrected data? Is this difference in bias coming from the original data or from the correction?

I believe the question that is being asked here is how the Proxy and Full Physics XCH4 products compare before and after bias correction. To compare this we can look at the TCCON validation for the data before bias correction. We believe that a full TCCON validation of non-bias-corrected data would be confusing for the reader, and unconstructive for the paper given the already lengthy TCCON validation. We therefore limit the discussion here to mean bias over all TCCON colocations.

Before bias correction the mean bias is 7.2 ppb and 13.3 ppb for the Full Physics and Proxy XCH4 products, respectively. These change to -0.1 ppb and -0.1 ppb after bias correction. So the average bias is fit to approximately zero. The RMSE is also slightly improved from bias correction, but the change is minor, about 0.5 ppb. Thus before bias correction, the Full Physics approach is closer to TCCON than the Proxy, as expected. We have added a discussion on this in the manuscript (L326). We have also included some additional information on how the bias correction performs on the Proxy and Full Physics products in section 3.4 (L176).

*L201 - Can you elaborate further how the value of*  $X_T$  *is decided upon?*

 $X_T$  is chosen to probe the steepest part of the curves in Figure 3 thus maximising the improvement that can be extracted from the machine learning filtering approach. We have added a statement on this to the manuscript (L278).

L211 - Does taking the different filtering approaches for land vs glint (L211) lead to significant differences in the sampling statistics? Could this lead to ocean/land biases in the final data? It would also be good to see how the data looks for a few single orbits that pass over both ocean/land.

In order to address the referee's points here, we have expanded section Appendix A in which ocean measurements are discussed, to include a description of the bias correction for glint mode, and added Figure A1 which shows 6 consecutive days of GOSAT-2 data - which is the revisit time of GOSAT-2. We also include a discussion on land/ocean biases (L486) revolving around the newly added Table A1, in which we quote the bias, per product, for TCCON stations that have colocated measurements over both land and ocean. Some stations used for validation of glint mode have very few colocations (1 or 2), so the significance of the statistics here should be taken into consideration when interpreting these conclusions. These stations are marked with an asterisk in Table A1. We find that the agreement between land/ocean is better for the Full Physics products, however the Proxy product shows some significant differences. This is explicitly stated in the manuscript (L491).

L217 – How true is that assumption and is it robust to issues such as sensor degradation (that we know GOSAT/GOSAT-2 can suffer from) which have a strong temporal component.

The machine learning models learn the relationship between a set of retrieval features and the quality of a measurement, which is defined as the difference with TCCON. Assuming that detector degradation leads to an increase in noise, and such an increase leads to a poorer quality retrieval, measurements suffering from such degradation should have a larger difference with TCCON and therefore be more likely to be flagged as bad quality. Looking at the feature importance analysis in Figure 2, the models across the different years are very similar, implying that this assumption is robust. We mention this briefly in the text (L242).

L237 – This may not be easily possible but it would be very interesting if the models for the different years themselves were all similar. They all clearly give consistent results but are the models compensating for different things in different years, to varying degrees. Some examination of the potential explainability of these models would be great but may not be possible.

To illustrate the explainability of the RFC quality filtering models, we take one of the products, in this case the Full Physics XCH4 product, and generate a feature importance plot that combines the different models across different years. This plot has now been added to Figure 2. We find that for each year, the feature importance is very consistent, with the order of features mostly the same across the different years, although there are small variations. The most important features are always either chi2, surface elevation variance, aerosol parameter or ratio of H2O column retrieved at different wavelengths, but the order can differ slightly depending on the year. We discuss the feature importance at L243.

L244 – This seems to be saying that as there's typically much less "bad" Proxy data, it is harder to identify the bad cases as they stand out less. Could you evaluate the model  $CO_2$  used in the proxy in a similar way here to separate the components?

The referee makes the correct interpretation here. TCCON validation of XCO2 used for the Proxy calculation would in principle be possible and may have the potential to more accurately train the filtering models for the Proxy product. It would be interesting to see if this can improve the performance metrics to bring these on par with the Full Physics models. Given that this is something experimental, we do not add this to the manuscript, but will investigate the result of this in considering a forthcoming version of the Proxy product.

L259 - Can you elaborate on why XCH₄ and XCO₂ FP (from the same retrieval?) have different yields?

This is because the increase in yield is with respect to the previous threshold filtering technique. Setting the values for these thresholds is rather subjective, and is quite sensitive to the data product. The arbitrarily chosen values for the CO2 product were less optimal than for the CH4 Full Physics product, leading to a larger improvement when applying the more objective RFC models. We clarify this point in L289.

L269 – Lots of mentions of TCCON prior to defining that it's GGG2020 that is used. I'd mention this sooner.

An extra reference to that fact that TCCON GGG2020 is used throughout has been added at L221.

Figure 3 – Am I correct that this mixes together land and glint data? Can you separate the two out for some sites to better understand any ocean/land bias in the data?

Figure 4 shows the TCCON validation for land retrievals only. We have added this detail to the caption and L305 states this as well. The results for TCCON validation of ocean retrievals are presented separately in Figure A2.

L304 – Is this a fair comparison between GOSAT and GOSAT-2 when you have applied this new post-filtering and hence can define/tune the RMSE for GOSAT-2?

The tunability of the RMSE is one of the strengths of the new QA value presented for GOSAT-2. It should be noted that the comparison here is between data products and we refrain from commenting on whether GOSAT or GOSAT-2 performs better as a mission. In that sense we believe the comparison is fair. To make this more clear, we have removed this paragraph from section 5.1 so that the TCCON validation section deals

only with GOSAT-2, and placed it at the end of section 5.2, where we compare the data products with their GOSAT counterparts. Here we also state that we are comparing data products of GOSAT and GOSAT-2 (L393).

L316 – Minor grammar issue "by in the ratio"

typo corrected

L337 - How are you matching GOSAT to GOSAT-2 data? i.e. What criteria do you use to find co-located soundings?

The data from the two satellites are matched by re-gridding XCH4 to 2x2 degree lat/lon boxes, per day. A colocation is considered when there are data from each satellite in the same grid cell for a given day. This has been added to the text (L363).

Figure 7 – Can you show similar maps for the other 2 products? Maybe in an appendix?

Similar maps of the XCO2 Full Physics and XCH4 Proxy products have been added in Figure C1 in a section in the Appendix. Reference is made to these figures at L369.

L381 – Can you outline how you match GOSAT-2 to TROPOMI?

The data from the two satellites are matched by re-gridding XCH4 to 2x2 degree lat/lon boxes, per day. A colocation is considered when there are data from each satellite in the same grid cell for a given day. We have added this to the text (L406). The colocation method is the same as for the GOSAT intercomparison, therefore to avoid repetition we refer to that part of the paper.

L387 – I don't quite understand this point about the bias remaining "effectively constant". The bias increases with QA value doesn't it? (i.e. from -4.6 to -6.3 ppb). Can you also comment on why the proxy bias seems to systematically decrease with increasing QA?

We thank the referee for this point, it has helped to clarify an important conclusion of the paper. We have decoupled the statement about the bias from the start of this paragraph as the key result is more that sigma\_TROPOMI follows sigma\_TCCON with QA value.

We have noted the systematic change in bias for the Full Physics product, however the change for the Proxy is 0.7 ppb. If 18 ppb is taken as 1 % of XCH4, this would correspond to 0.04 %, a change of which we are not sensitive to, so interpretation of this should be taken with caution. Both of these points have been added to the manuscript (L424).

L390 – typo (missing word after Northern?)

This indeed was a mistake in the wording and has now been fixed.

Response to RC2 – Gregory Osterman

Line 31: anomaly is misspelle. maybe "increase" instead of "excess"

Line 31: typo corrected and "excess" changed to "increase"

Line 43: classified instead of "classed"

Line 43: "classed" changed to "classified"

Lines 53 and 60: Authors mention level-1B and level 2 without really saying what that means, though they do mention radiance spectra. Maybe that could be clarified?

We have slightly reordered the paragraph that introduces GOSAT to expand the definition of L1B data. Also the referee is correct, since reference is repeatedly made to level 2 data throughout the paper, this deserves its own definition. We give this at Line 62.

Line 65: Could you provide a little bit more information it the text as to why "Science with TANSO-FTS-2 has been limited, and more restricted to total column products". You have the references, but it could be helpful to provide a brief mention of why that is the case so readers do not need to go the references for context.

Line 68: Several interesting science studies are worth noting here, therefore we have decided to rework this paragraph. Particularly the very recently submitted work of Janardanan et al. comparing flux inversions of CH4, when assimilating GOSAT or GOSAT-2 XCH4 data, and the retrieval of HDO/H2O ration from combined NIR and TIR bands. We also introduce here the level 2 products of NIES and IUP-Bremen, rather than do this in section 5.1, as it makes the flow of the discussion better.

Line 70: "has for long been widespread used"

Line 77: phrase changed

Line 73: TCCON uses different methods of obtaining vertical profiles at their sites, maybe mention balloon-borne observations (AirCore) in addition to aircraft. Also maybe mention that these are site specific and can vary quite a bit.

Line 80: reference to AirCore added along with Karion et al. 2010.

Lines 258-265: The authors highlight the results from the RFC model of quality controlling the GOSAT-2 data. Figure 2 includes single data points showing what data filtering using table 1 would look like. But that is not mentioned in the text. What would the results for ocean data look like from table 1?

Reference to data filtering using Table 1 in comparison to the RFC filtering was made already (see Line 286). For Figure 3, the number of collocated measurements over ocean is so small compared to land (factor of 100), that a data point for ocean would look like it lies along the x-axis. We therefore made no update to the manuscript.

Figure 3: The authors mention on line 276 that "These are single soundings of GOSAT-2 over land compared to an average of the TCCON measurements that coincide spatially and temporally", is that the reason for the lines of (along the x-axis) of data that appear in the scatter plot. What would a daily average of both satellite and TCCON look like? Would there be much difference?

Figure 4: the referee is correct, lines of data in the x-axis direction result from taking the daily averaged TCCON value plotted against single soundings of GOSAT-2, and are indicative of a bias over geolocation. A daily average of GOSAT-2 data would reduce the overall scatter considerably. GCOS however define the precision in terms of single soundings. We address this point in Line 308.

Line 309: Why are there a disproportionately higher number of collocations at Caltech, Edwards and Xianghe? Maybe a little more explanation on that?

At Line 340, we have expanded a bit more this discussion to explain why there are more colocations at Edwards and Caltech, and the impact of this on the results.

Section 5.2: I was not clear on this, maybe I missed it, but did you use the RCF on the GOSAT data (I think you did not). How is the GOSAT data quality filtered? Is it similar to the Table 1 approach? What changes with the comparison if you use Table 1 and compare to GOSAT?

There are two points made by the reviewer here 1) How are the GOSAT data quality filtered and 2) how do the results change when comparing GOSAT and GOSAT-2 both filtered with Table 1. We shall answer these both respectively. i) Indeed the RFC filtering has not been applied on GOSAT data. The GOSAT data are filtered according to

Table 1. The referee here raises an important point that has been missed. We have therefore included a sentence at the start of section 5.2 (Line 364) to highlight how the filtering of GOSAT is done. ii) If GOSAT-2 is filtered according to Table 1 (therefore the same as GOSAT) the bias is reduced for the Full Physics XCH4 product, which goes from -15 ppb to -9.3 ppb. For the Proxy XCH4 product the effect is less where the bias changes from -5.3 ppb to -6.5 ppb. To avoid confusion for the reader as to which results are presented for which quality filtering, we do not include these values in the manuscript.

Line 375/Table 4: Again, I apologize if I missed this, but do you recalculate the TCCON comparisons to include data that has GOSAT-2/TROPOMI comparisons? Table 4 are the global numbers for different QA values. Column 6 is the scatter for a TCCON comparison on just the GOSAT-2/TROPOMI subset? That number is higher than Table 3?

The results presented in Table 4 are for the TCCON validation for all GOSAT-2/TCCON colocations. This is now explicitly stated in Line 423. The referee makes a good point here that Table 4 should be consistent with Table 3. This was not the case due to a different way of defining sigma\_TCCON. To avoid confusion we have recalculated the values for sigma\_TCCON in Table 4 as the mean RMSE over all station, as done in Table 3. We also state this in Line 424.

Line 390-392: The wording here is confusing, maybe rephrase this to make clear that there is a large difference over North Africa and that they could be attributed to high aerosol levels from dust and burning events causing issues for the retrievals.

Line 427: There was a mistake made with this sentence. It has been reworded

Line 396: "The global average is close to zero", I think you mean that the aggregate global difference between TROPOMI and GOSAT-2 is close to zero?

Line 437: Wording has been clarified

Could this data filtering model be used on GOSAT? Would that make the data sets from the two satellites more compatible?

The RFC filtering could indeed be applied to GOSAT, and this would make for a better comparison allowing one to more concretely comment on the performance of GOSAT vs GOSAT-2. We consider this out of the scope of the paper, which presents the GOSAT-2 data products, and would be more suitable for a follow-up paper. We address this point in Line 395 of the manuscript.

**List of revisions:**

- Definitions of L1b and L2 data at lines 55 and 63 respectively.
- Rework of introductory paragraph on GOSAT-2 covering several interesting science studies leveraging the capacities of GOSAT-2 at line 72.
- Reference to different calibrations used at TCCON stations at lines 84/85.
- Addition of Figure 1 showing spectral fits and residual for a typical high quality measurement from the Physics retrieval, discussed at line 125.
- Clarification of retrieved information on ILS at line 160.
- Elaboration on the bias correction and how the Physics retrieval performs compared to the Proxy at line 182.
- Inclusion of aerosol parameter, w, to Table 1 which was missed out by mistake, with definition at line 221, and highlighting of filters not applicable to the Proxy product by an asterisk in Table 1.
- Expansion of Figure 2 to include two new panels. Panel (a) shows histograms of various geophysical parameters in the high quality training dataset and their values. This is discussed at line 232. Panel (c) shows a feature importance analysis of the XCH4 Full Physics product for the different models across the five years. This is discussed at line 248.
- Clarification on how XT is chosen at line 284.
- Clarification of why CO2 and CH4 product improvement is different at line 295.
- Explanation of the reason behind lines of data in the x-direction in Figure 4, at line 314.
- Comparison of the TCCON validation results of the non-bias corrected for the XCH4 Full Physics and Proxy products at line
- Section 5.1: paragraph comparing TCCON validation for GOSAT and GOSAT-2 has been moved to the end of section 5.2 which discusses the comparison between GOSAT and GOSAT-2.
- Elaboration of why Caltech and Edwards have a disproportionally larger number of colocations with GOSAT-2 than other TCCON sites at line 352.
- Introduction of GOSAT-2 level 2 products from NIES and Noel et al. (2021, 2022) has been moved to section 1 where GOSAT-2 is introduced.
- Clarification of how GOSAT and GOSAT-2 data are collocated at line 382.
- Line 387/Appendix C: Reference to maps of GOSAT and GOSAT-2 for the XCH4 Proxy and XCO2 Full Physics products which have been newly added in Figure C1.
- Clarification of how TROPOMI and GOSAT-2 data are collocated at line 428.
- Table 4 definition of sigma\_TCCON changed to match the one used in Table 3.
- Line 446 Clarification of which TCCON/GOSAT-2 colocations are used to calculate the statistics of N\_TCCON and sigma\_TCCON in Table 4.

- Line 448 Correction that there is systematic change in bias between GOSAT-2 XCH4 FP and TROPOMI, while change for XCH4 PR is negligible.
- Additional statistics of the bias between TROPOMI and GOSAT-2 are given at line 456 for cases where the albedo is greater than 0.4 only.
- Figure A1/Line 495: new figure plotting single overpasses of GOSAT-2 to show XCH4 over land and ocean simultaneously.
- Explanation of the bias correction for ocean measurements at line 496.
- Table A1: New table which compares TCCON validation results for land and ocean. TCCON sites are considered if they have collocated data with GOSAT-2 over both land and ocean.
- Discussion of land/ocean biases for the 3 GOSAT-2 products at line 511.
- Table B1: new column which gives the extent of the total available timeseries at each TCCON station.
- Appendix C: new section to present global distribution of GOSAT compared to GOSAT-2, for the XCH4 Proxy and XCO2 Full Physics products.